# Whole Class or Small Group Fluency Instruction: A Tutorial of Four Effective Approaches

**Melanie R. Kuhn**

Department of Curriculum and Instruction, Purdue University, West Lafayette, IN 47906, USA; melaniek@purdue.edu

**Abstract:** Four scientifically validated approaches to fluency instruction (Fluency-Oriented Reading Instruction, Wide Fluency-Oriented Reading Instruction, Fluency-Oriented Oral Reading, and Wide Fluency-Oriented Oral Reading) are reviewed. Two for the whole class and two for small groups. Key components of fluency, automaticity, and prosody are defined, and their contribution to reading comprehension is discussed. Automaticity contributes through its freeing up of attention to attend to meaning, and prosody contributes through its addressing of pacing and expression that, in turn, reflect textual meaning. Four principles for effective fluency instruction are also presented: Modeling, extensive opportunities for practice, the use of scaffolding, and the incorporation of prosodic elements. The four instructional approaches presented in this article are based on two different strategies for integrating extensive opportunities to read: Repeated versus wide reading. All four approaches use challenging texts, or texts at the upper end of the learners' zone of proximal development, thus providing learners with access to a broader range of vocabulary and concepts than would be the case if they read only instructional level texts. All four also provided highly effective procedures for either whole-class or small-group reading instruction. The goal of this summary is to provide readers with effective approaches for classroom instruction.

**Keywords:** fluency; repeated reading; wide reading; challenging texts

---

When thinking about reading development, word recognition and comprehension are generally at the fore (e.g., [1,2]). However, since the publication of the National Reading Panel (NRP) report [3], fluency has received an increased amount of attention as a link between the two. As research indicates (e.g., [4,5]), fluency is critical to the overall reading process, and it is important to discuss not only the components of fluency—automaticity and prosody—but also how they come together in fluent reading. This article will briefly consider fluency's role in terms of development and then present a review of four research-based instructional approaches, two for the whole class and two a for small group, that have been shown to increase students' word recognition, fluency, and comprehension. The goal of the review is to provide a concise summary of these highly effective approaches. Importantly, they are also easy-to-implement additions to the reading instruction that occurs in the classroom. Given the challenges many children face in their reading development and the importance of scaffolded reading in developing students' reading, these approaches can be an important addition to your reading curriculum.

## 1. Fluency's Role in Reading Development

According to researchers (e.g., [6,7]), fluency development is important for two reasons. First, it helps learners shift from slow and deliberate word decoding to automatic word recognition. Second, it allows readers to apply prosodic (e.g., expressive) elements to text. For most learners, this stage of development occurs after they have established basic decoding skills in the first grade. Traditionally, fluency instruction begins in the latter part of first grade and continues throughout third

grade (e.g., [8]), even as learners are developing their knowledge of more complex word patterns (e.g., long vowel sounds, diagraphs, etc.). Further, effective reading is built upon strong oral language, and the two reinforce one another (e.g., strong oral language helps students develop their phonemic awareness, which, in turn, helps learners develop alphabetic knowledge) [3].

The primary goals of fluency instruction are two-fold, to help students consolidate what they have learned about word recognition and to apply elements of oral language to text (e.g., [4,5]). In terms of the former, the shift is important because it allows readers to attend to the meaning of what is being read rather than focusing their attention on word identification [6]. Prosody also contributes to fluency through the application of elements of oral language to what is being read [9], and it provides a unique contribution to comprehension [10]. However, fluent reading does not develop simply by teaching word recognition in isolation. Instead, it needs to be combined with the extensive scaffolded reading of connected text (e.g., [4]).

Automaticity. Beginning, or novice [11], readers spend a great deal of attention, decoding each word they encounter in a text. In fact, when you listen to first graders read aloud, you will likely notice that their reading is uneven and lacking in expression. Because they expend so much of their attention on word recognition, they have little left over to determine the text's meaning. While this type of decoding is typical of beginning readers, it is important that learners consolidate what they have learned about word recognition as they shift toward reading fluency. In other words, they need to develop automaticity, so their word recognition is quick, accurate, and effortless. This shift frees up their attention, allowing them to comprehend what is being read.

Prosody. Components of prosody, such as phrasing, stress, and emphasis, all combine to make a unique contribution to comprehension beyond that of automatic word recognition [10]. As skilled readers, when we read a text, these elements work together to reflect the meaning of the text, thereby enhancing comprehension. Luckily, some of these elements can be represented by punctuation. However, several aspects of prosody cannot. For example, it is usually clear where phrasing exists in spoken language, but, while some written phrases are identified through commas, others are implied. If students are to comprehend fully, they need to learn how to apply these elements to what is being read. This can be accomplished through instruction that focuses on phrasing rather than simply on increasing learners' reading rate [12].

Principles of fluency instruction. Before discussing specific approaches to fluency instruction, it is useful to outline those instructional elements that effective approaches have in common (e.g., [13,14]). These are: Modeling, extensive opportunities for practice, the use of scaffolding, and the incorporation of prosodic elements. As learners shift from intentional decoding to fluent reading, the primary instructional focus needs to be on helping students read connected texts. This practice can occur either through guided instruction (assisted reading) or through independent practice (unassisted reading) [15]. Such practice allows students to apply their developing knowledge of both word identification and prosody in context. By focusing on connected text, students are also learning to construct meaning and to recognize when their comprehension breaks down [11].

The first principle of fluency instruction is to model what fluent reading should sound like. This allows students to develop a clear understanding of their goal (e.g., [13,14]). This can be accomplished by reading aloud as part of the classroom routine, and the genre used should range from poetry to non-fiction. Unfortunately, while such modeling is important, it will not assist learners in developing their own fluency in and of itself. Instead, the second principle notes that, for students to become fluent readers, they need to spend a substantial amount of time actually reading. One of the key components of the approaches discussed below is that they maximize the amount of time students spend reading connected text, increasing the average number of minutes from under 10 per day (e.g., [16]) to between 20 and 30 (e.g., [17]).

The third principle of effective fluency instruction is the incorporation of support or scaffolding (e.g., [13,14]). To maximize fluency instruction's effectiveness, it needs to make use of challenging texts (i.e., those at the higher end of students' zone of proximal development). Scaffolding is



necessary if students are to succeed with material that would otherwise be too difficult (e.g., [4]). Both unassisted approaches and assisted approaches make use of some form of scaffolding [9]. Unassisted approaches involve repeated readings in which students independently read a passage multiple times. This repetition serves as the reader's scaffold. In assisted approaches, students read along with the teacher or another more-skilled reader who provides them with support. Assisted approaches can be used either with repeated readings (i.e., a single text that is read multiple times) or with wide reading (i.e., multiple texts that are read one time each for a similar amount of time). Without this support, students are unlikely to be successful when reading challenging material.

The final principle involves integrating prosodic elements into fluency instruction. It is important to address prosody explicitly so that students learn to focus on appropriate pacing, along with other elements that contribute to expression, rather than simply on speed [9,12]. When considering a particular fluency procedure, it is helpful to check the strategy against these principles to ensure its effectiveness and to maximize the instructional time available. The next section will discuss four research-based instructional approaches (two whole class and two small group interventions) that do just that. The whole class approaches are meant for second and third-grade classrooms. It is useful to think about these approaches in terms of the Common Core State Standards or the standards for individual states. Not only are they effective at developing students' fluency, they are useful in covering a range of challenging texts including those assigned for language arts, social studies, and science. The small-group procedures can be used with disfluent readers at any grade level.

## 2. Fluency Oriented Reading Instruction (FORI) and Wide FORI

When considering effective approaches for your whole class, FORI and Wide FORI have an extensive research base. Both are designed for the primary or shared reading selection for a given week. However, FORI uses a single text over five days, whereas Wide FORI makes use of three selections [17]. After reading through the description and discussion, you can decide which is a better fit for you.

### 2.1. Fluency-Oriented Reading Instruction (FORI)

FORI [18] was originally designed in response to a district mandate that all classroom instruction be conducted using grade-level texts. Teachers recognized that this would be problematic for their struggling readers, so they teamed up with a researcher to develop a lesson plan around these selections. Since they did not want to set students up for failure, they decided to integrate scaffolding along with repetition to increase their students' access to the text. While the teachers in the initial implementation used a core reading program, the approach has been replicated using literature anthologies as well as trade books (e.g., [17]). Notably, the second graders who participated in the initial study saw an average growth of 1.8 years in the first year and 1.7 years in the second year [18] according to results on the Qualitative Reading Inventory [19].

The FORI lesson plan is based upon an easy to implement a five-day schedule (see Table 1). However, there are some important provisos regarding the lessons. First, one of the most important goals is to increase the amount of time students spend reading connected text. Given that students need to read between 20 and 30 min, short selections should not be used. Instead, books or stories that are typical in length for second or third grade are most appropriate. Next, when teachers fail to pay attention to implementing the procedure in a purposeful way, it ceases to be effective. This is especially important given the structure of the approach which children enjoy, but which can feel monotonous to teachers. However, it is also important to bear in mind that the procedure allows students to read challenging material and make accelerated progress, so it is worth enduring a little teacher tedium. On the plus side, the regular structure can reduce the pressure to create new lesson plans each week, allowing teachers to focus on vocabulary and comprehension development instead. The day-by-day lesson plan is presented below.

**Table 1.** FORI lesson plan.

| Monday | Tuesday | Wednesday | Thursday | Friday |
|---|---|---|---|---|
| - The teacher uses pre-reading activities to introduce the selection to the class.<br>- The teacher then reads the selection to the class, students should follow along on their own copies.<br>- The teacher and students discuss the selection, thereby keeping the focus on comprehension. | - The teacher and students echo read the selection.<br>- The teacher should integrate comprehension and vocabulary strategies throughout the reading.<br>- Optional:<br>Discussion of the selection can continue. | - The teacher and students choral read the selection. | - The students partner read the selection while the teacher monitors and supports their reading. | - The students complete extension activities designed to broaden their understanding of the text. |
| Time required: approximately 40 min | Time required: approximately 30 min | Time required: approximately 20 min | Time required: approximately 30 min | Time required: 30–40 min |

DAY 1—Introducing the Text. The primary goal for the first day is to introduce the text that the students will be reading over the course of the week. This usually begins with the selection of a pre-reading activity and may include making predictions, discussing critical vocabulary, or building background knowledge. For example, if you were reading about Mae Jemison, you might introduce terms like astronaut and space shuttle, talk about some of the work that occurs at NASA, or ask your students to think about the kinds of challenges faced by individuals who explore outer space. The instruction should also be designed to build student interest and motivation.

The next step involves reading the selection aloud to the class. At this point, it is critical that the students follow along in their own copies of the text. One way to help ensure the students are on-task is to circulate around the room, making sure they are following along. Your fluent reading of the selection is important for two reasons. In terms of mechanics, it provides students with a model of what their reading should sound like and allows them to look at the words as they are being pronounced without having to decode them independently upon initially encountering them. Second, it allows them to listen to the entire text before they have to read it, keeping the focus on meaning. The read-aloud should be followed with a discussion that will help the students grapple with their understanding of the selection and reinforce the notion that comprehension is the primary goal of reading.

DAY 2—Echo Reading. The second day involves an echo reading of the selection. This procedure involves reading a section of text and having the class read it back. If the class is not familiar with this procedure, start with one or two sentences at a time. While students will eventually be able to read it back independently, it is helpful to read along with them as they develop this ability. Next, as students get used to the concept of echo reading, it is important to move them toward longer sections of text. A paragraph or two is usually a good length since it is too long for students to rely on their short-term memory, but not so long that they will lose track of what they are reading. Eventually, however, they may be able to echo read as much as a page at a time, although that will depend on the layout of the book and the amount of text on a page. As is the case on the other days, it is helpful to circulate around the room to help students stay on task. Depending on the complexity of the material, Day 2 can also have a focus on comprehension. This can prevent students from perceiving the purpose of the echo reading as simply word identification. Any comprehension activity that focuses on the meaning of the text, from interspersing questions throughout the reading to some form of discussion to a written response, would be appropriate (e.g., [20]).

DAY 3—Choral Reading. On Day 3, the students and teacher read aloud the selection simultaneously or chorally. Since this is the third time the text is being read, it is particularly important to maintain your energy in terms of reading fluidly and expressively. As was the case on the previous days, it also continues to be important to circulate around the room to help students maintain their focus and pacing. However, as they get more acquainted with the lesson format,

they will improve their ability to attend to the reading. Since the third reading takes the least amount of time, there should be plenty of time remaining for literacy activities that extend beyond the week's primary selection.

DAY 4—Partner Reading. On Day 4, the students work in pairs to read the selection for the final time. Partner reading is saved for last because it provides for the least amount of support, so the students need to have established a degree of comfort with the text, something which should have occurred over the previous three days. When it comes to determining pairs, the two best approaches are self-selection, or allowing the students to choose who they want to read with themselves, and assigning partners across ability levels [21]. When assigning readers, it is important for a difference in achievement to exist between the two readers, but that difference should not be too great. If the abilities are too disparate, the more or the less skilled reader (or both) is likely to experience frustration with the process.

The simplest approach to rank readers is to list students from the highest achieving reader to the student experiencing the greatest difficulty (e.g., [22]). You should then divide the list in half, and place the first name from the second list next to the first name on the first list and so forth (see Table 2). This ensures that, in terms of reading development, your most skilled reader is partnered with a reader from the middle of the class, and the reader who is having the greatest difficulty will also be partnered with an average reader—keeping the difference between students relatively equal for all pairs.

**Table 2.** Grid for determining partners.

| Skill Rank | Name | Skill Rank | Name |
|---|---|---|---|
| 1 | Bianca | 11 | Nichelle |
| 2 | Theo | 12 | Lilly |
| 3 | Miguel | 13 | Erica |
| 4 | Asia | 14 | David |
| 5 | Thad | 15 | Sydney |
| 6 | Maxwell | 16 | Lee |
| 7 | Hazel | 17 | AJ |
| 8 | Xander | 18 | Marco |
| 9 | Krissie | 19 | James |
| 10 | Bernard | 20 | Topher |

In terms of the reading itself, each partner should read alternating pages or paragraphs. Since the students have already read this material at least twice (and followed along on an additional reading), they should be able to provide one another support and coaching if either experiences difficulty. However, they should feel free to ask for help as needed. When a reader gets to the end of a page, they should finish any sentence or paragraph even if it continues onto the next page. The partner should then take over the reading of the selection. If partners complete the first reading of the text and there is enough time, they can switch assigned pages and read it for a second time.

DAY 5-Extension Activities. Day 5 is set aside for extension activities that can be used to develop a deeper understanding of the text. This is important for developing skills such as student-led discussions, writing responses, constructing charts and diagrams, or any other approach that helps learners better comprehend what they have read. Since students will need to refer back to the selection to complete many of these activities, this can be a powerful opportunity to teach them the type of skills required by the Common Core State Standards (CCSS) [23] or the literacy standards used in a particular state.

## 2.2. Wide Fluency Oriented-Reading Instruction (Wide FORI)

Wide Fluency-Oriented Reading Instruction or Wide FORI [17] differs from FORI in terms of the number of texts that students will read in a five-day lesson plan (three as opposed to one; see Table 3). Despite this difference, the two approaches follow similar formats, are both easy-to-implement, are designed to ensure students spend an extensive amount of time reading connected text, emphasize comprehension, and have a solid research-base. Moreover, because both approaches use challenging texts, students are introduced to a variety of concepts and vocabulary that would not be accessible if they were limited to reading instructional level texts.

When thinking about these approaches, it is important to consider what happens when students either read a single text repeatedly or read several selections for an equivalent amount of time. Since both procedures involve reading connected texts, students encounter an extensive number of words, phrases, and concepts—allowing them to develop better fluency and improved comprehension [24]. Having already discussed FORI, the day-by-day lesson plan for Wide FORI is presented below.

**Table 3.** Wide FORI lesson plan.

| Monday | Tuesday | Wednesday | Thursday | Friday |
|---|---|---|---|---|
| Text #1<br><br>- The teacher uses pre-reading activities to introduce the selection to the class.<br>- The teacher then reads the selection to the class, students should follow along on their own copies.<br>- The teacher and students discuss the selection, thereby keeping the focus on comprehension. | Text #1<br><br>- The teacher and students echo read the selection.<br>- The teacher should integrate comprehension and vocabulary strategies throughout the reading.<br>- Optional:<br><br>Discussion of the selection can continue. | Text #1<br><br>- The students complete extension activities designed to deepen their understanding of the text. | Text #2<br><br>- The teacher and students echo read a second selection for the week.<br>- The teacher should integrate comprehension and vocabulary strategies throughout the reading.<br>- Discuss selection. | Text #3<br><br>- The teacher and students echo read a third selection for the week.<br>- The teacher should integrate comprehension and vocabulary strategies throughout the reading.<br>- Discuss selection. |
| Time required: approximately 40 min | Time required: approximately 30 min | Time required: 30–40 min | Time required: approximately 30 min | Time required: approximately 30 min |

DAY 1—Introducing the Text. The first two days of Wide FORI parallel the first two days of the FORI procedure. After selecting the primary text for the week, you begin by introducing it with your typical pre-reading lesson. These can include vocabulary development, building background knowledge, and making predictions, they should also help to build interest and motivation for reading. The next step involves reading the text aloud to the students as they follow along in their own copies. As is the case with Day 1 of FORI, this allows them to hear an expressive, fluent reading of the text without having to decode the words the first time they encounter them. It is also important to circulate around the room as a way to monitor the students and redirect those who may need redirection. After completing the reading, a discussion should take both to help students develop a deeper understanding of the selection and to emphasize that the construction of meaning, not word recognition, is the central goal.

DAY 2—Echo Reading. Day 2 of the Wide FORI lesson plan again parallels Day 2 of the FORI procedure to a large degree. The lesson starts with an echo reading of the text that was introduced on Day 1 (see Day 2 of FORI for details on implementing the echo reading procedure). To maintain a focus on meaning, comprehension strategies should also be used during or following this second reading of the material. Discussion, questioning, and summarization are good examples of the type of activity that can occur at this point. The difference between the two approaches occurs at the end of the lesson, in that a partner reading of the text can take place if time permits (see Table X.2 for an example of how students can be partnered).

DAY 3—Extension Activities. The first major divergence from the FORI format occurs on Day 3. Rather than undertaking an additional reading of the selection, extension activities are scheduled for this day. Again, any activity designed to help students delve deeper into the text can be used (e.g., graphic organizers, discussions, or written responses). Given less time is spent on this material, it is critical that students have a solid understanding of what they read by the end of the third day, even if it means devoting a slightly greater proportion of your literacy block to its comprehension.

DAYS 4 and 5—Echo Reading. The fourth and fifth days incorporate a second and third text. Since an echo-reading of each text is required as part of the lesson, there is a limited amount of time available to spend on the pre- and post-reading activities. Given these time constraints, it is important to consider exactly what additional instruction should be used and how to best extend students' understanding of what they read. This can best be accomplished by carefully matching activities, such as vocabulary development, discussion, or summarization, to the instructional goal. As with Day 2, if there is time available on Days 4 or 5, the students can partner read that day's selection.

## 3. Fluency-Oriented Oral Reading (FOOR) and Wide Fluency Oriented Oral Reading (Wide FOOR)

Should you be working with small groups of students who need additional fluency support when compared to their peers, Fluency-Oriented Oral Reading (FOOR) and Wide Fluency-Oriented Oral Reading (Wide FOOR) are two research-based approaches that integrate the principles discussed earlier in this paper [25]. These interventions were designed to further examine the finding that it is the reading of connected text, rather than the repetition per se, which ensures readers become fluent [4]. In other words, students who read for a given amount of time, whether by using multiple selections or by repeatedly reading a single selection, made equivalent gains in automaticity and prosody. These approaches differ from FORI and Wide-FORI in that they are designed for small groups of struggling readers rather than for a class as a whole. As such, they are perfect for either small-group work or as a Tier II intervention.

The research intervention was originally designed for four groups of five or six second graders. These groups met three times a week for 15–20 min per session, and all of the students were identified as struggling readers by their teachers. This assessment was confirmed when the Test of Word Recognition Efficiency (TOWRE) [26] and the Qualitative Reading Inventory-II [19] were used as pre-tests (the same measures were also used as post-tests. The Fluency-Oriented Oral Reading group (FOOR) echo or choral read a single title three times over the course of a week (see Table 4), whereas the Wide Fluency-Oriented Oral Reading group (Wide FOOR) echo or choral read three different titles during their weekly lessons (see Table 5) [25]. A third group listened to a fluent reading of the Wide FOOR titles, and a fourth group only participated in the pre- and post-testing (i.e., they did not receive any additional reading instruction beyond that already taking place in their literacy curriculum). The texts selected were considered to range from late first grade to early third grade levels according to Fountas and Pinnell's ratings [27], and titles such as *Hooray for the Golly Sisters* [28], *The Case of the Cat's Meow* [29], and *Whistle for Willie* [30] were included in the study. While these books were considered challenging for the students, they were able to read them successfully since the instruction was heavily scaffolded.

**Table 4.** Fluency-Oriented Oral Reading (FOOR) lesson plan.

| Monday (approximately 20 min) | Wednesday (approximately 20 min) | Friday (approximately 20 min) |
|---|---|---|
| - Echo read text for the first time<br>- Comprehension activities should be included, but brief, especially on the first day | - Choral read the material (if the students need additional support, echo read instead) | - Students partner read the selection (if the students finish early, they can begin a second partner reading, even if they are not able to complete it) |

**Table 5.** Wide FOOR lesson plan.

| Monday (around 20 min) | Wednesday (around 20 min) | Friday (around 20 min) |
| --- | --- | --- |
| Echo read the **first** text. Briefly discuss the material as part of the lesson. | Echo read the **second** text. Briefly discuss the material as part of the lesson | Echo read the **third** text. Briefly discuss the material as part of the lesson |

The study yielded important results that can be used for small group fluency instruction [25]. When the students were post-tested on the TOWRE and the QRI-II, students in both the FOOR and Wide FOOR groups scored better than the other students in terms of word recognition in isolation, prosody, and the number of correct words read per minute (automaticity) in connected text. However, the students in the Wide FOOR group made greater gains in comprehension than did their peers who were in the FOOR group. It seems that this outcome may be reflective of the types of reading the FOOR and the Wide FOOR groups undertook. Since repetition was used during the FOOR intervention, the students may have viewed the re-reading process as a way to improve their word recognition and prosody, in other words, they might have focused on their accuracy and how they sounded when reading aloud. Since the students in the Wide FOOR group read a different book at each session, they are more likely to have concentrated on making sense of what they were reading, rather than just on how their oral reading sounded. While these differences are implicit, researchers have found similar results in other interventions [31,32]. In fact, it seems that simply asking students to focus on the meaning of the material they are reading results in increased comprehension. As such, it is important to remind readers that they should focus on the meaning of the text, even when they are learning to decode or trying to develop their fluency [11].

Given the above findings, there are several takeaways for small group fluency instruction [25]. First, whether you base your intervention on repetition or wide reading, it is essential that students spend a significant amount of time reading connected text. Each FOOR and Wide FOOR session incorporates 15–20 min of reading in addition to the reading instruction that was already occurring in the classroom. Second, the text being read needs to be challenging for the students. This does not mean the students should feel frustrated, but they need to be reading material that is between 85%–90% accuracy since they are receiving significant support. Further, it is important to re-evaluate the reading material every few weeks to determine how well students are reading and to move them to more challenging selections when the texts become too easy. By evaluating student learning in this way, it will be easier to determine when students no longer need additional fluency support. At that point, they will more likely benefit from alternative forms of instruction that focus on other key competencies of reading. It is also important to remember that, while this approach was used with groups of five-to-six students, it can be used in smaller groups, with pairs of students, or even in a one-to-one tutorial format.

### 3.1. Fluency-Oriented Oral Reading (FOOR)

DAY 1–Introducing the Text. The primary purpose of the first day is to echo read the selection (see the section on FORI and Wide FORI for the echo reading protocol). Since time is limited (15–20 min), comprehension instruction should be brief. It can involve a short introduction, be embedded in the reading, or involve a post-reading discussion, but it should not be the primary focus of the lesson or the main use of the time available. On the other hand, as was noted above, it is especially important that students who participate in a repeated reading intervention are reminded that understanding is the primary goal of reading [25].

DAY 2–Choral Reading. The second day incorporates a choral reading of the selection, however, if the students need more support with a particular selection, then it is reasonable to echo read the text a second time. Since the students should be somewhat familiar with the material, this is a good time to include questions during the reading, clarify vocabulary, or incorporate a discussion. Again, this confirms the idea that understanding is the goal of reading, while keeping fluency at the fore. However,

the essential aspect of the lesson is the re-reading of the selection, so this should take precedence in terms of time.

DAY 3–Partner Reading. The third reading of the text occurs on Day 3. Students should partner read the selection since this requires them to take responsibility for the entire text (See the section on FORI for ways to create and implement partnerships). Since there will be a maximum of three pairs in any group, it should be relatively easy to monitor the students and assist those who are having difficulties. If there is enough time, the students can re-read the text one more time, preferably reading the pages they listened to the first time through. If there is an uneven number of students, the teacher or another adult should act as a partner.

### 3.2. Wide Fluency-Oriented Oral Reading (Wide FOOR)

As is the case with FOOR, the time per session is limited (between 15–20 min) [25]. However, since you will be reading a new selection each time you meet, it is even more important that you complete the echo reading within the time allotted (see the FORI section for a discussion of echo reading). As a result, any comprehension instruction should be relatively brief. However, predictions, vocabulary, and questions strategically interspersed within the selection are all appropriate choices, as are many others. Since a different text is read at each session, this approach lends itself especially well to a post-reading discussion—as long as the reading is completed. It is also critical to remember that, as students become increasingly fluent, it is necessary to increase the level of challenge. This process should continue until the students are reading grade-level material comfortably. Once students have achieved this milestone, you should consider whether additional fluency instruction should be undertaken or whether the time would be better spent on other focal areas of reading instruction.

Since the students will be reading a new text each time they meet, it is useful to think about whether the weekly material should be connected in some way (e.g., author, subject, or theme). While this is not necessary, it can serve to strengthen students' vocabulary and conceptual knowledge (e.g., [12,33]). Another issue involves finding enough appropriate material. Thankfully, there are usually texts available in the school building or via the Internet. For example, sets of books can be gathered from a guided reading program, selections can be taken from commercial core programs that are no longer being used, and copies of unused trade books can often be found in storage. In addition, student magazines and child-friendly websites often have substantial articles on subjects that are of interest to young learners. The critical elements to bear in mind when choosing texts are that the material should always be challenging for the students, every student must have their own copy, and what is being read needs to be engaging for students.

### 3.3. Future Directions for Research

When considering the above instructional approaches, it is important to note that they were all used with second graders, either for whole class instruction (FORI, Wide FORI) [17] or with small groups of struggling readers (FOOR, Wide FOOR) [25]. This does not mean they will not work with other learners—there is a substantial research base indicating similar approaches are effective with older struggling readers (e.g., [4,14]). However, it is also important to note that there is a need for research to confirm the effectiveness of these approaches with other groups of learners. Similarly, the selections being used were either exclusively (FOOR, Wide FOOR) [25] or predominately (FORI, Wide FORI) [17] fiction materials that were either part of the literacy curriculum or requested by the teachers. As such, future directions for research include determining the effectiveness of these approaches for other grades and with other types of texts. Results from any adaptation of the lesson plans will help determine their usefulness beyond the examples presented here.

### 4. Conclusions

When considering whether to use a whole class approach to fluency development or an approach designed for smaller groups of disfluent readers, it is important to consider students' developmental

needs [8,11]. For example, it is rare that an entire class of first graders will be ready for fluency instruction. Similarly, most students beyond the third grade should be fairly fluent readers. In both of these scenarios, small group instruction using FOOR or Wide FOOR is appropriate for those learners mentioned, but is unlikely to benefit the entire class. In fact, these approaches are efficient and effective ways to meet various students' needs as they develop and change throughout the school year. By using a flexible grouping structure, you can vary your curriculum so that oral reading instruction is targeted to those who need it the most.

On the other hand, it is the case that most, if not all, second and third graders could benefit from the type of whole-class fluency instruction provided by FORI and Wide FORI. These approaches have a solid scientific research base, have been proven to increase students' fluency and overall reading achievement, and are an effective way to teach the shared reading component of a literacy block. However, it is also important to remember that even the best oral reading instruction should not be the entirety of a literacy curriculum. Further, once students are fluent readers, these approaches should be afforded less time and priority in the curriculum, or even phased out entirely. Instead, additional silent reading and comprehension instruction, writing activities, and/or opportunities to expand your students' reading of different genres and in different content areas should take precedence. In this way, it becomes possible to create literacy instruction that will meet the needs of all students.

**Funding:** This research was supported in part by the Interagency Education Research Initiative, a program of research jointly managed by the National Science Foundation, the Institute of Education Sciences in the U.S. Department of Education, and the National Institute of Child Health and Human Development in the National Institutes of Health (NIH Grant 5 R01 HD40746-4).

**Acknowledgments:** I would like to acknowledge the work of the grant team, especially Paula Schwanenflugel and Steven Stahl who served as PI for the for the grant, Lesley Morrow, Elizabeth Meisinger, Barbara Bradley, Robin Morris, Rose Sevcik, and Deborah Woo who made integral contributions to the research, and the many graduate students who worked at our multiple sites across the country.

**Conflicts of Interest:** The author declares no conflict of interest.

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
