# Peer review of "Whole Class or Small Group Fluency Instruction: A Tutorial of Four Effective Approaches"

_education, doi:10.3390/educsci10050145_

Round 1

Reviewer 1 Report

This is a very good paper, I have not suggestions for improvement.

Author Response

Thank you for taking the time to review my paper.  It is much appreciated.

Reviewer 2 Report

The topic of the paper is interesting and the description of the ways the approaches were implemented in class is explicit. However, it is not clear what was the impact of the adoption of the four approaches on the students' reading performance in the four settings. Nor is it clear how the reading materials have been selected or the criteria that have been adopted to guide the selection of the reading materials. To make things worse, it is not clear how the participating students' current reading abilities were measured or determined. In other words, it is not clear how the four approaches have been 'scientifically validated'.

Author Response

Thank you for taking the time to review my paper.  I am responding to your concerns below:

However, it is not clear what was the impact of the adoption of the four approaches on the students' reading performance in the four settings. Nor is it clear how the reading materials have been selected or the criteria that have been adopted to guide the selection of the reading materials. To make things worse, it is not clear how the participating students' current reading abilities were measured or determined. In other words, it is not clear how the four approaches have been 'scientifically validated'.

Your points are valid. However, this piece was invited to be an instructional piece that provides implementation guidance. The research described here has been reviewed in multiple places previously; this piece serves as an overview and is meant to improve the quality of instruction, especially since all four approaches are easy to implement and far more effective than much of the teaching that currently occurs in may reading classrooms.  The original research follows scientifically valid research methods, and all your questions have been addressed in the earlier published papers. I will make sure it is clear that this is a review rather than a new research study.

Again, thank you for your efforts on behalf of this work.

Reviewer 3 Report

Overall, this paper is well organized and discusses a topic of major importance.

Title:  The reviewer feels this title could be more informative to a reader.  It appears that this might be a paper that is a tutorial based on research rather than a traditional research paper.  If so, it might help the reader to include something like “A tutorial for whole class and small group fluency instruction”

Background: 

Although the background provides important content, it could be improved by clearly stating the rationale for this paper.  The reviewer was confused on whether this was a tutorial or a research study.  Additionally, it might be helpful to posit the need for improved reading instruction as there is a low rate of reading proficiency amongst children in elementary school.

The background might be improved by adding a reading theory such as Gough and Tunmer or Scarborough’s reading braid. 

Additionally, it may be helpful to put in Common Core State Standards for 2nd and 3rd graders. 

Lines 35-40 might benefit in underscoring specifically why oral language helps reading.  In other words, reading comprehension requires good oral language skills in addition to word recognition (Gough & Tunmer).  It may be important to further expand on line 40 where oral language has a unique contribution to comprehension.

Again, this does not seem like a research study.  There were no research questions posted.  Is this a tutorial?

Whole Classroom FORI and Wide FORI

These sections are well written and easy to understand.  However, upon reading other sections, was this based on a research study by authors?  Were there improvements in any outcome measures? Or is this just a tutorial?

Small Group FOOR AND Wide FORI

Line 253 talks of a research intervention but there really isn’t a good description of participants for procedures.  How long was this study?  What type of second graders were these?

Line 268 The study yielded important results. Were these results statistically significant? Are there effect sizes?  “scored better” does not mean much.

Were improvements seen in children?  If so, how long were these interventions implemented before?

Author Response

Thank you for taking the time to review my paper.  I have responded to your points below:

The reviewer feels this title could be more informative to a reader. It appears that this might be a paper that is a tutorial based on research rather than a traditional research paper. If so, it might help the reader to include something like “A tutorial for whole class and small group fluency instruction”

This is an excellent point and I made your suggested change.

Although the background provides important content, it could be improved by clearly stating the rationale for this paper. The reviewer was confused on whether this was a tutorial or a research study. Additionally, it might be helpful to posit the need for improved reading instruction as there is a low rate of reading proficiency amongst children in elementary school.

I have attempted to make this clearer.

The background might be improved by adding a reading theory such as Gough and Tunmer or Scarborough’s reading braid. Additionally, it may be helpful to put in Common Core State Standards for 2nd and 3rd graders. 

While I consider myself to be solidly in the research-based reading "camp", I do not consider the Simple View of Reading an adequate model and, therefore, would prefer not to include it as a resource. However, I feel there is adequate theory provided and have attempted to highlight it. I have also incorporated a footnote regarding the CCSS and other state standards and mentioned the connection to fluency, the use of challenging books, and connection to content areas.

Lines 35-40 might benefit in underscoring specifically why oral language helps reading. In other words, reading comprehension requires good oral language skills in addition to word recognition (Gough & Tunmer). It may be important to further expand on line 40 where oral language has a unique contribution to comprehension.

I attempted to address this using the NRP and provided an example.

As for the remaining issues, I clarified that this is a review of effective instructional interventions rather than a study per se. The original small group study involved a small number of students so an analysis using nonparametric stats was implemented. However, the findings developed from it have been replicated elsewhere, including the larger intervention discussed in the paper and multiple small group interventions that followed the same format. 

Overall, I appreciate your comments and the time you took to review this paper.  I am certain that addressing your points will make for a stronger article.

Round 2

Reviewer 2 Report

Thanks for the clarification that "this piece was invited to be an instructional piece that provides implementation guidance". In this case, this is a kind of action research and the reader should be alerted to this special purpose at the very beginning and thus may have a proper expectation for the paper. 

My suggestion is, therefore, that the author explicitly advises the reader on the purpose of proving implementational models for FORI and FOOR in the initial part of the paper and the reasons for the choice of reading materials "either exclusively or predominately fiction" and also what the action research aims to address or solve via the action reported in the paper.

A minor error of misprint (FOOR) in the subtitle "Fluency-Oriented Oral Reading (FOOR) and Wide Fluency Oriented Oral Reading (FORI)" needs to be fixed on Line 249.

Author Response

Thanks for the clarification that "this piece was invited to be an instructional piece that provides implementation guidance". In this case, this is a kind of action research and the reader should be alerted to this special purpose at the very beginning and thus may have a proper expectation for the paper.

My suggestion is, therefore, that the author explicitly advises the reader on the purpose of proving implementational models for FORI and FOOR in the initial part of the paper and the reasons for the choice of reading materials "either exclusively or predominately fiction" and also what the action research aims to address or solve via the action reported in the paper.

A minor error of misprint (FOOR) in the subtitle "Fluency-Oriented Oral Reading (FOOR) and Wide Fluency Oriented Oral Reading (FORI)" needs to be fixed on Line 249.

I added a sentence about the purpose of the review in the abstract as well as clarified the goals in the introduction.  Since the material used was part of the classrooms' literacy material or selections requested by teachers, that has been noted.  I also made the change in the subtitle.

Reviewer 3 Report

Thank you for addressing the comments.

Author Response

Thank you